# Internet of Things-Based Automated Solutions Utilizing Machine Learning for Smart and Real-Time Irrigation Management: A Review

**DOI:** 10.3390/s24237480

**Published:** 2024-11-23

**Authors:** Bryan Nsoh, Abia Katimbo, Hongzhi Guo, Derek M. Heeren, Hope Njuki Nakabuye, Xin Qiao, Yufeng Ge, Daran R. Rudnick, Joshua Wanyama, Erion Bwambale, Shafik Kiraga

**Affiliations:** 1Department of Biological Systems Engineering, University of Nebraska-Lincoln, Lincoln, NE 68588, USA; 2West Central Research, Extension, and Education Center, University of Nebraska-Lincoln, North Platte, NE 69101, USA; 3School of Computing, University of Nebraska-Lincoln, Lincoln, NE 68588, USA; 4Texas A&M AgriLife, 1102 East Drew Street, Lubbock, TX 79403, USA; hopenjuki.nakabuye@ag.tamu.edu; 5Panhandle Research, Extension, and Education Center, University of Nebraska-Lincoln, Scottsbluff, NE 69361, USA; 6Carl and Melinda Helwig Department of Biological and Agricultural Engineering, Kansas State University, Manhattan, KS 66506, USA; 7Department of Agricultural and Biosystems Engineering, Makerere University, Kampala P.O. Box 7062, Uganda; wanyama2002@gmail.com (J.W.);; 8Center for Precision and Automated Agricultural Systems, Irrigated Agriculture Research and Extension Center, Department of Biological Systems Engineering, Washington State University, Prosser, WA 99350, USA

**Keywords:** precision irrigation, sensor networks, artificial intelligence, precision agriculture, water use efficiency, crop productivity, remote monitoring, smart farming, edge computing, interoperability

## Abstract

This systematic review critically evaluates the current state and future potential of real-time, end-to-end smart, and automated irrigation management systems, focusing on integrating the Internet of Things (IoTs) and machine learning technologies for enhanced agricultural water use efficiency and crop productivity. In this review, the automation of each component is examined in the irrigation management pipeline from data collection to application while analyzing its effectiveness, efficiency, and integration with various precision agriculture technologies. It also investigates the role of the interoperability, standardization, and cybersecurity of IoT-based automated solutions for irrigation applications. Furthermore, in this review, the existing gaps are identified and solutions are proposed for seamless integration across multiple sensor suites for automated systems, aiming to achieve fully autonomous and scalable irrigation management. The findings highlight the transformative potential of automated irrigation systems to address global food challenges by optimizing water use and maximizing crop yields.

## 1. Introduction

The challenge of feeding a growing population with finite resources is becoming increasingly pressing. By 2050, the world population is expected to reach 9.7 billion, necessitating a 70% increase in food production [1]. Irrigation plays a crucial role in enhancing crop yields and agricultural productivity to meet this growing demand. Studies have shown that irrigation can significantly increase crop water productivity, contributing to increased food production [2,3]. However, water scarcity poses a significant challenge, with many regions facing water deficits and the need for improved water management practices [4]. Optimizing irrigation schedules and doses based on crop water requirements and environmental conditions like evaporative demand is essential for maximizing the yield and quality while minimizing water use [5]. The necessity of scalable water-efficient practices for increasing food demand cannot be overstated. Techniques such as regulated deficit irrigation, the use of treated wastewater for irrigation, and the cultivation of drought-tolerant crops like sorghum and cotton have shown promise in improving water productivity and ensuring food security [6,7]. As the global food challenge intensifies, it is imperative to critically evaluate the current state and future potential of irrigation management systems to guide the research, innovation, and implementation efforts towards fully autonomous and scalable solutions.

Despite the importance of irrigation in addressing the global food challenge, traditional irrigation management techniques have significant limitations. These techniques include manual scheduling, which relies on crop observation, soil feel, and intuition; timer-based scheduling, where irrigation occurs at predetermined intervals regardless of actual crop needs; and sensor-based methods using ET modeling and soil moisture sensing. Manual and timer-based scheduling can lead to high operational costs and inefficient water use [8]. The reliance on manual intervention and predetermined schedules limits their adaptability to changing environmental conditions, crop water requirements, and soil moisture levels [9]. Sensor-based irrigation systems offer an alternative, enabling real-time adjustments based on soil water status measurements [9]. However, the adoption of these systems in commercial settings has been limited, often requiring extensive input from researchers [10]. The limitations of traditional irrigation management techniques highlight the need for scalable, automated solutions for greater efficiency in irrigation management.

Automated systems that collect real-time data, analyze it, and make autonomous irrigation decisions can lead to improved water use efficiency, increased crop productivity, and significant energy savings [8,11]. The recent studies have demonstrated that smart irrigation systems utilizing IoT and machine learning technologies can reduce energy consumption by 30–40% compared to traditional methods by optimizing pump operation and water delivery schedules [12,13]. These energy savings are particularly significant given rising energy costs in agricultural operations, making automated systems increasingly attractive for both environmental and economic sustainability [14]. The optimization of irrigation scheduling and water delivery not only conserves water resources but also minimizes the energy required for pumping and distribution, leading to reduced operational costs while maintaining or improving crop yields.

The emergence of smart irrigation management and the Internet of Things (IoT) marks a significant shift from historical irrigation practices. Smart irrigation management involves using advanced technologies and data-driven approaches to optimize water usage and improve crop yield. IoT, which refers to the interconnection of devices and sensors through the internet, enables the collection and exchange of vast amounts of data in real-time. Modern approaches rely on vast data and analysis algorithms, leveraging technologies such as remote sensing, sensor networks, weather data, and computational algorithms [15]. The IoT plays a vital role in collecting vast amounts of data through sensors, data transmission, and tailored networks, enabling real-time monitoring and control of irrigation systems [16]. These advancements in data collection and analysis have the potential to revolutionize irrigation management, allowing for more precise and efficient water use. However, challenges such as processing diverse data sources, data integration, and the lack of integrated data analysis hamper the full benefits of IoT in irrigation management [17]. The current fragmented approach in smart irrigation, focusing on individual components rather than the entire system, limits the potential for fully autonomous, real-time end-to-end irrigation management [15]. To address these challenges and fully realize the potential of smart irrigation management, there is a need for automating and integrating each section of the irrigation management pipeline, from sensor/weather data collection and transmission to processing, analysis, decision-making, and automated action. This integration requires a thorough investigation of the role of interoperability and standardization in enabling communication and compatibility between components within the automated irrigation management pipeline.

On the other hand, machine learning (ML) plays a significant role in processing vast amounts of data, predicting plant water stress, modeling climate effects, and optimizing irrigation in smart irrigation management systems. ML algorithms, such as artificial neural networks and support vector machines, can analyze data collected from sensors and weather stations to determine optimal irrigation schedules based on patterns and insights gleaned from the data [18]. For example, ML models can predict crop water requirements based on historical weather data, soil moisture levels, and plant growth stages, enabling more precise and timely irrigation decisions. However, the potential of ML is often constrained by manual steps, such as data interpretation, decision making on irrigation timing and volume, and system adjustments. Automating ML integration to allow direct action from insights to irrigation execution, removing bottlenecks, and achieving real-time adaptability is crucial for fully autonomous irrigation management. By integrating ML into automated systems, the irrigation management pipeline can become more efficient, enabling real-time decision-making and action based on data-driven insights. To achieve this level of automation and integration, it is essential to identify gaps and propose solutions for integration across the automated irrigation management system, aiming to achieve fully autonomous and scalable irrigation management.

To achieve integration across the automated irrigation management system, interoperability and standardization are critical. Interoperability allows different system components, such as sensors, actuators, and software, to communicate and exchange data effectively, while standardization ensures that data are represented in a consistent format [19,20,21]. The existing and emerging standards such as the Open Geospatial Consortium (OGC) SensorThings API [22] and ISO 11783 (tractors and machinery for agriculture and forestry—serial control and communications data network) [23] have applicability to real-time irrigation management systems. The OGC SensorThings API provides a standardized way of connecting IoT devices, managing and retrieving sensor data, and interacting with data using RESTful web services. The ISO 11783 standard series, for instance, defines protocols and messages for agricultural machinery communication, providing a foundation for interoperable irrigation systems [24]. These standards contribute to the integration of automated irrigation management systems by enabling seamless communication and data exchange between various components, such as sensors, controllers, and decision support systems. However, challenges such as data quality, scalability, reliability, and security need to be addressed to fully realize the potential of interoperability and standardization in automated irrigation management systems. Addressing these challenges is crucial for enabling the integration of components within the automated irrigation management pipeline, which is essential for achieving fully autonomous, scalable irrigation management. A comprehensive evaluation of the interoperability and standardization’s role in this integration is necessary to guide future research and implementation efforts.

The primary objective of this systematic review is to critically evaluate the current state and future potential of real-time, end-to-end automated irrigation management systems that integrate IoT and machine learning technologies for enhancing agricultural water use efficiency and crop productivity. The focus on these objectives is crucial because fully automated, integrated irrigation management systems have the potential to significantly improve water use efficiency and crop yields, which are essential for addressing the global food challenge. By examining the automation of each part of the irrigation management pipeline and the seamless integration of each section, this review aims to identify the current state of technology and the gaps that need to be addressed to achieve fully autonomous irrigation management. Analyzing the effectiveness and efficiency of integrated end-to-end automated irrigation systems will provide valuable insights into the benefits and limitations of these systems, guiding the future research and development efforts. Investigating the role of interoperability and standardization is essential for understanding how to enable the integration of components within the automated irrigation management pipeline, which is crucial for achieving scalable and reliable systems. Finally, by identifying gaps and proposing solutions for integration across the automated irrigation management system, this review aims to provide a roadmap for future research and implementation efforts towards fully autonomous, scalable irrigation management that can contribute to addressing the global food challenge.

## 2. Review Methodology

In this review, a comprehensive, multi-stage methodology is employed to identify, evaluate, and synthesize the research on real-time, autonomous irrigation management systems. The review process, shown in Figure 1, permits a detailed analysis of the current developments in this field.

The literature search encompassed multiple scientific databases, including Google Scholar, Scopus, and Web of Science, using a targeted set of search terms such as ‘smart irrigation’, ‘IoT irrigation’, ‘automated irrigation’, and ‘precision irrigation’. The search included academic research papers, relevant standards and guidelines from various organizations, and studies from adjacent fields that could inform irrigation management practices. This broader scope captured the latest scientific findings, and established best practices, industry benchmarks, and insights from related domains such as precision agriculture, remote sensing, machine learning, and IoT technologies.

The literature review began with a database keyword search yielding 1485 potential papers. Title and abstract screening using predefined criteria reduced this to 462 papers for full-text review. Stringent evaluation focusing on detailed system architectures, data processing, control algorithms, and performance metrics initially selected 123 papers. Throughout this review, additional relevant works were identified via supplementary searches, citations, and expert recommendations. These underwent the same rigorous evaluation, with valuable additions incorporated. This iterative process culminated in a final selection of 166 papers, standards, and resources, ensuring a comprehensive, up-to-date review while maintaining high quality and relevance standards.

A question-driven framework, centered on the key research questions and hypotheses (Table 1), guided the analysis of the selected literature. These questions addressed the overarching objective of critically evaluating the current state and future potential of real-time, end-to-end automated irrigation management systems while considering contributions and implications from adjacent fields such as networking, machine learning, cybersecurity and high-performance computing and the role of industry standards and guidelines.

## 3. Data Collection to Cloud: Automation and Real-Time Processing

### 3.1. Data Collection to Cloud: IoT Architecture for Smart Irrigation

The automation of data collection through Internet of Things (IoT) technologies is a critical component in developing real-time, end-to-end automated irrigation management systems. IoT devices and structures enable the efficient gathering of vital information about crop health, environmental conditions, and water requirements, essential for enhancing agricultural water use efficiency and crop productivity. In this section, how IoT architectures are implemented in practical irrigation systems is examined, with specific field implementations further detailed in the case studies in Section 7. The evolution of these systems from basic monitoring to fully automated control demonstrates the practical realization of IoT capabilities in agricultural settings.

As shown in Figure 2, data collection is the first step in the IoT-based architecture for smart and automated irrigation management. The success of these systems depends on the efficient collection, transmission, and analysis of various data types. Among these, soil moisture data are arguably the most critical, as it directly reflects water availability to plants and informs optimal irrigation timing and amount [25,26]. Soil moisture sensors, such as tensiometers, capacitance probes, and time-domain reflectometry (TDR) sensors, provide real-time measurements of soil water content at various depths [27]. These measurements enable the calculation of soil water stress indices, like the Soil Water Stress Index (SWSI) [28,29], and the observation of soil moisture depletion trends. SWSI quantifies the level of water stress experienced by plants based on available soil moisture relative to critical thresholds such as field capacity and permanent wilting point [30]. Such indices provide a normalized measure of soil water stress, potentially ranging from 0 (no stress) to 1 (maximum stress), which can be used to inform precise irrigation scheduling, preventing both over- and under-watering [25].

In addition to soil moisture, canopy temperature data, collected using infrared thermometers and thermal cameras, are valuable for assessing plant water stress and adjusting irrigation [31]. These data can be used to calculate indices like the Crop Water Stress Index (CWSI), which compares actual canopy-air temperature differences to theoretical wet and dry references [32,33,34]. However, the interpretation and application of thermal data present significant challenges [35], such as issues in utilizing thermal remote sensing images for crop and soil surface temperature information. While thermal sensors provide reliable spatial and temporal trends, accurate absolute canopy temperature measurements remain challenging, with one study showing a strong correlation (R^2^ = 0.89) between low-cost and professional thermal sensors [36]. Moreover, crop stress detection using thermal data is highly time-sensitive, with pre-irrigation thermal images under cloud-free conditions better reflecting water stress than post-irrigation images [37]. These complexities complicate the integration of thermal data into real-time irrigation management systems. Nevertheless, thermal stress indices may be more effective for deficit-irrigated crops, as demonstrated by Shellie and King [38], who showed that using daily CWSI thresholds for irrigation scheduling in deficit-irrigated grapevines reduced water use without compromising yield or berry composition.

To further enhance the irrigation decision-making, advanced machine learning algorithms can be applied to weather data to predict crop evapotranspiration (ET) under specific conditions [33,39]. By integrating these diverse data types and derived indices, IoT-based smart irrigation systems enable real-time monitoring and data-driven decision-making. Continuously analyzing sensor data and calculating various stress indices allows these systems to provide timely irrigation recommendations, aiming to reduce water waste and optimize crop yield. Furthermore, the historical data collected can be used to refine irrigation strategies over time, adapting to specific crop needs and local environmental conditions. This integration of diverse data types and derived indices represents a promising approach to advancing precision agriculture and enabling more efficient and sustainable water management practices.

To ensure the effectiveness of these smart irrigation systems, the continuous monitoring of field conditions and the irrigation infrastructure is crucial. A significant challenge in implementing such systems is the limited availability of reliable internet connectivity in rural agricultural settings, which can be a major barrier to the adoption of advanced irrigation technologies. The recent research has addressed this challenge through systems designed specifically for low-connectivity environments. Modern automated irrigation systems can maintain core functionality during internet outages through local control features and edge computing, with systems capable of making autonomous irrigation decisions based on soil moisture sensor readings even without cloud connectivity [40]. These systems can operate in automatic mode by utilizing historical weather data for rain prediction and maintaining operation through scheduled irrigation even during connectivity issues [41].

Within this framework, IoT sensor networks, computer vision, and multispectral and hyperspectral imaging provide real-time data on soil moisture, temperature, humidity, and crop health [42]. The efficiency of data transmission and access to information is facilitated by edge computing and local processing capabilities, allowing systems to function independently when internet access is limited. This approach enables farmers to maintain reliable irrigation control even in areas with poor connectivity, with systems designed to store and process data locally until network access is restored [40]. Additionally, novel techniques such as entropy-based measures and data visualization offer richer insights into the overall system’s health [43].

However, sensor errors and drift can impact the data accuracy and reliability. To mitigate these issues, best practices for sensor calibration and maintenance, such as regular cleaning, reference measurements, and calibration curves, are essential for ensuring data quality [44]. Advanced techniques for data quality control and assurance, like outlier detection, data interpolation, and sensor drift compensation, can be applied in real-time irrigation monitoring and decision support systems [45]. Moreover, data standards and interoperability, such as SensorML [46], AgGateway’s ADAPT (ANSI/ASABE S632 [47]) framework, and OGC’s SensorThings API, enable effective monitoring and ensure system reliability [46]. Metadata and data provenance, aligned with relevant standards (e.g., ISO 19115-1:2014 [48]), are also essential for ensuring the traceability, interpretability, and trustworthiness of irrigation monitoring data [49,50].

#### 3.1.1. IoT Devices and Automation of Data Collection

The automation of data collection through Internet of Things (IoT) technologies is a critical component in developing real-time, end-to-end automated irrigation management systems. IoT devices and structures enable the efficient gathering of vital information about crop health, environmental conditions, and water requirements, essential for enhancing agricultural water use efficiency and crop productivity. IoT devices commonly used in irrigation management include soil moisture sensors, weather stations, and flow meters. These devices are often integrated into wireless sensor networks (WSNs) with energy-efficient communication protocols, offering a cost-effective and scalable solution for real-time data collection in large-scale irrigation systems [51]. For instance, [52] demonstrated an automated irrigation control system using WSNs that efficiently manages water in cultivated fields, improving network lifetime and reducing water consumption.

Advanced sensing technologies, such as hyperspectral imaging and thermal sensing, have emerged as powerful tools for non-invasive plant stress detection within IoT frameworks. These technologies provide valuable information about crop traits, enabling the early and accurate detection of plant health issues [53]. In a study by Guzinski et al. [54], a modified TSEB model was used to highlight the importance of high-resolution data in accurately capturing the spatial and temporal dynamics of energy fluxes influenced by environmental factors. Recent advances in satellite technology have significantly improved remote sensing capabilities for irrigation management. Commercial providers now offer daily satellite imagery at a 3-m resolution, suitable for smallholder irrigation applications [55]. This development has influenced the balance between satellite and UAV-based solutions. The irrigation management industry is exploring solutions based on satellite and weather data, potentially reducing reliance on ground truth sensors in some applications [56]. Satellite-based irrigation management solutions have shown agreement with ground-based measurements in recent studies [57]. Despite these advances, IoT-enabled tools such as unmanned aerial vehicles (UAVs) can still offer higher spatial and temporal resolution in specific scenarios, providing a more granular perspective on crop health and development [58,59]. Zhang et al. [60] exemplifies the potential of this approach, demonstrating the use of hyperspectral remote sensing and machine learning to predict leaf water content in rice plants, offering valuable information for optimizing irrigation strategies and improving water use efficiency.

The integration of IoT devices with cloud computing and edge processing has shown potential to enhance the capabilities of automated irrigation systems. For example, Premkumar and Sigappi [61] proposed an IoT-enabled edge computing model for smart irrigation systems, which reported improved performance in terms of latency, bandwidth, throughput, and response time compared to cloud only solutions in their specific experimental setup. These IoT-based systems aim to improve data transmission reliability in challenging environments and may extend the lifespan of sensor nodes. They also offer remote monitoring and automated control capabilities, potentially allowing for more timely adjustments to irrigation schedules based on current field conditions [62].

#### 3.1.2. Data Types and Formats

When collecting and utilizing data for automated irrigation systems, factors such as volume, frequency, format, and source are relevant [27]. Figure 3 illustrates the primary data types utilized in automated irrigation systems, along with their respective sources and collection methods. This visualization emphasizes the diverse range of information integrated to optimize irrigation management, while clarifying the relationships between data types, their origins, and acquisition techniques. The volume of data generated can be substantial, necessitating efficient data storage, processing, and transmission technologies [63]. Data collection frequency impacts the temporal resolution and the ability to detect rapid changes in plant water status or environmental conditions. Camilo and Beckwith [64] demonstrated that collecting full pulse resolution data from water meters provided more accurate estimates of event occurrence and features compared to aggregated temporal resolutions. Data format determines the compatibility and interoperability with analysis tools and platforms [65]. Standardized data formats, such as JSON, XML, or CSV, facilitate data exchange and integration between different components of the automated irrigation system [65]. Data sources can impact reliability, accuracy, and spatial coverage. In-field sensors provide localized measurements, while remote sensing platforms offer data at larger spatial scales [31]. Combining data from multiple sources allows for a more comprehensive understanding of crop water requirements and optimizes irrigation management. Data quality, accuracy, and reliability are paramount in irrigation management, as they directly impact decision-making and water use efficiency [66]. Researchers have investigated data compression, aggregation, and filtering techniques to reduce bandwidth requirements and improve transmission efficiency [67]. Data standardization and harmonization are also crucial for facilitating integration and interoperability between system components. Metadata provide context and enable better data interpretation and decision-making [68]. By incorporating metadata into the data management and analysis pipeline, decision-makers can make more informed and context-aware decisions, leading to improved water use efficiency and crop productivity [68].

#### 3.1.3. Data Quality and Preprocessing

Maintaining high data quality is paramount for automated irrigation systems. The initial steps of data processing involve cleaning, preprocessing, and fusing data from various sources to ensure accuracy and reliability. Table 2 provides an overview of the key techniques employed in these processes, highlighting their benefits and contributions to the overall system performance. By addressing challenges such as missing values, inconsistencies, outliers, normalization, and data fusion, these techniques improve data reliability, leading to more informed and accurate irrigation decisions.

### 3.2. Data Transmission: Edge Computing and Fog Computing

Edge computing and fog computing are being explored as technologies to enhance real-time irrigation management, with the aim of improving efficiency, scalability, and reliability. However, agricultural IoT systems face several critical data processing bottlenecks that must be addressed. Recent analysis has identified specific constraints including data transmission latency due to network congestion, processing power limitations at local gateways, and challenges in real-time analysis of massive sensor datasets [75]. These bottlenecks particularly impact time-sensitive irrigation decisions, where delayed processing can lead to suboptimal water management.

Implementation of hybrid edge-cloud architectures has demonstrated promising solutions to these processing challenges. Studies combining edge computing with efficient data management algorithms have achieved significant improvements, including a 30% reduction in data processing time and a 25% improvement in real-time monitoring capabilities through optimized local processing and data transmission [75]. More recent advances using TinyML algorithms at the edge have shown even more dramatic improvements, with processing times decreasing fourfold when scaling from 1 to 10 processing nodes while maintaining 88–95% compression efficiency [76]. These performance gains are particularly valuable for irrigation management, where the rapid processing of sensor data directly impacts the timeliness and effectiveness of water application decisions.

Looking to the future, emerging technologies like federated learning present possibilities for further addressing processing bottlenecks. This approach enables distributed processing between edge devices and the cloud while minimizing data transfer, addressing both efficiency and privacy concerns in IoT networks [76]. However, the challenges remain related to the heterogeneous and volatile nature of devices in edge and fog computing tiers, underscoring the ongoing nature of research in this area.

These transmission capabilities form the foundation for the real-time operation demonstrated in the case studies (Section 7), where implementations achieve practical control of irrigation systems through timely data collection and processing. The specific architectures and protocols discussed here enable the performance characteristics detailed in those implementations.

#### 3.2.1. Real-Time Data Transmission Protocols and Technologies

Real-time data transmission is critical for automated irrigation management systems as it enables the timely delivery of sensor data to the cloud for processing and decision-making. The message queuing telemetry transport (MQTT) protocol has emerged as a popular choice for real-time data transmission in IoT networks due to its lightweight nature, publish–subscribe architecture, and suitability for low-bandwidth networks [77]. Client-server IoT networks, such as those based on MQTT, play a crucial role in the real-time data transmission for automated irrigation management systems [78]. This architecture enables efficient data collection, processing, and dissemination, facilitating the integration of various components within the automated irrigation management pipeline. Other application layer protocols, such as XMPP [79], CoAP [80], SOAP [81], and HTTP [82], have also been explored for real-time data transmission in IoT networks, each with its strengths and weaknesses depending on the specific application and scenario [83].

#### 3.2.2. Challenges and Solutions in Real-Time Data Transmission for IoT-Based Irrigation

Real-time data transmission in automated irrigation management systems faces significant challenges due to environmental factors, technical limitations, and data security and privacy concerns. Agricultural environments are particularly challenging due to adverse weather conditions such as heavy rain, fog, and strong winds, which can attenuate or block radio signals, leading to data loss [84]. Dense vegetation and uneven terrain further complicate signal propagation [85]. Additionally, the limited range of traditional wireless technologies and network congestion caused by numerous sensors transmitting concurrently can lead to delays and data loss, hindering real-time decision-making.

To address these challenges, researchers have explored various advanced networking technologies. Cognitive radio networks (CRNs) and dynamic spectrum access (DSA) are promising solutions for optimizing spectrum utilization and reducing interference. CRNs enable devices to sense and adapt to the surrounding radio environment, dynamically adjusting transmission parameters to avoid interference [86]. DSA complements CRN technology by dynamically allocating unused spectrum, enhancing spectrum utilization, and reducing congestion. Integrating CRNs and DSA into the IoT network architecture requires careful consideration of spectrum sensing techniques, network topology, and data security [86].

Furthermore, implementing robust and adaptive communication protocols such as LoRa or ZigBee with suitable range and penetration capabilities is crucial for overcoming the interference and signal degradation caused by weather conditions and physical obstacles. Techniques like frequency hopping and error correction codes improve communication resilience and mitigate data loss. Optimizing network architecture, including deploying a distributed network with edge and fog computing capabilities, enhances data processing and transmission efficiency. Data optimization techniques such as compression, aggregation, and filtering are also vital in improving transmission efficiency.

Lastly, data security and privacy are paramount concerns, necessitating robust security measures to prevent unauthorized access and data breaches [66]. Effectively addressing these challenges requires a multifaceted approach that combines advanced networking technologies, robust communication protocols, and comprehensive security measures.

## 4. Data Storage and Processing: Automated Data Processing and Storage

Effective data processing is crucial for automated irrigation management systems to ensure optimal decision-making and water usage efficiency. In this section, various techniques and strategies employed in the cloud for handling data quality, preprocessing, scalable deployment, model inference, and online learning are outlined. These approaches are fundamental in transforming raw sensor data into actionable insights, enabling real-time adjustments to irrigation schedules based on current conditions.

### 4.1. Scalable and Autonomous Deployment Using Containerization Strategies

Scalable deployment is a critical aspect of cloud-based data processing in automated irrigation systems. Containerization, a method of packaging software code and all its dependencies so that it can run uniformly and consistently on any infrastructure [87], plays a crucial role in this process. Technologies like Docker and Kubernetes facilitate the efficient deployment and scaling of data processing modules in irrigation management systems.

In the context of automated irrigation, containerization allows for the encapsulation of various software components such as data processing algorithms, irrigation scheduling modules, and weather prediction models into separate, portable units. For example, Gualpa et al. [88] demonstrated the use of docker containers in a smart irrigation system, where each container housed specific software functionalities like data analysis and control mechanisms. This approach enables easy updates and modifications to individual components without affecting the entire system. Figure 4 depicts such a containerization management system.

The scalable process in containerized irrigation systems involves the following:packaging irrigation algorithms and data processing modules into containers;deploying these containers across multiple servers or cloud instances;automatically scaling the number of container instances based on data volume or processing needs;ensuring consistent performance across different environments, from edge computing devices to cloud servers.

### 4.2. Containerization Strategies

Table 3 summarizes the key containerization technologies and their benefits, explaining how they enable responsive and adaptive data processing pipelines for irrigation management. These technologies support the efficient resource utilization, isolation, and portability of applications, crucial for maintaining system performance and reliability. For instance, Kubernetes can automatically adjust the number of running container instances based on the incoming data volume from field sensors, ensuring optimal resource allocation during peak irrigation periods or when processing large volumes of weather forecast data.

By leveraging containerization, irrigation management systems can achieve greater flexibility, scalability, and reliability. This approach allows for the rapid deployment of software updates, the easy integration of new algorithms or data sources, and consistent performance across diverse agricultural environments. Containerization also facilitates the implementation of microservices architecture, where each containerized component can be developed, updated, and scaled independently, leading to more agile and maintainable irrigation management systems.

### 4.3. Irrigation Decision Making with Machine Learning Models

Machine learning (ML) models optimize irrigation management through tasks like soil moisture prediction and crop evapotranspiration estimation. However, challenges persist in dataset quality and model transferability. Shimim et al. [93] partially addressed this by integrating satellite imagery and infield sensors for evapotranspiration prediction (RMSE: 0.11–0.31 mm d^−1^), but noted limitations in obtaining fine-resolution spatiotemporal data.

The adaptability of ML models across different agricultural environments presents both challenges and opportunities. Recent research has demonstrated that spatiotemporal modeling approaches can achieve consistent performance across varying conditions, with prediction accuracies of 90% for soil moisture content across different soil depths and types when integrating multiple environmental variables [94]. This adaptability can be further enhanced through robust decision-making frameworks that explicitly account for environmental uncertainties. For instance, ensemble approaches combining random forests, support vector machines, and neural networks have maintained high performance (R^2^ > 0.87) across different environmental conditions by leveraging the complementary strengths of each method [95]. These advances in model adaptation are particularly valuable for irrigation management, where environmental conditions can vary significantly both spatially and temporally. The key to improving model adaptability lies in comprehensive environmental monitoring, continuous model updating based on new observations, and careful validation across different agricultural contexts. This approach ensures that ML models can maintain reliable performance while adapting to the diverse conditions encountered in real-world agricultural settings.

Despite these challenges, progress is being made. Dolaptsis et al. [96] demonstrated the effectiveness of long short-term memory (LSTM) networks in irrigation scheduling for maize crops. Their hybrid LSTM approach, incorporating data from soil sensors, weather stations, and satellite imagery, achieved high accuracy in predicting soil moisture content reduction (R^2^ values: 0.7602–0.9181) across different field conditions. This application showcases the potential of ML models to integrate diverse data sources and provide precise irrigation recommendations, while highlighting ongoing efforts to address transferability issues.

In the realm of crop evapotranspiration prediction, Egipto et al. [39] compared various ML algorithms, including support vector machines (SVMs) and random forests, for estimating actual crop evapotranspiration under non-standard conditions in vineyards. Their study found that ML approaches outperformed traditional methods, with the best-performing models achieving R^2^ values exceeding 0.89, demonstrating the superior accuracy of ML in capturing complex relationships between environmental variables and crop water requirements. To address the dynamic nature of agricultural environments, online learning algorithms enable the continuous update and improvement of ML models based on incoming real-time data. These algorithms are particularly valuable in irrigation management, where conditions can change rapidly due to weather patterns, soil moisture fluctuations, and crop growth stages. For example, Gelete and Yaseen [97] developed a hybrid model combining deep learning, nonlinear system identification, and ensemble tree intelligence algorithms for pan-evaporation estimation. Their approach, which incorporated principles of online learning, showed significant improvements in prediction accuracy compared to single models, with reductions in root mean square error (RMSE) ranging from 32.505% to 62.615% during the validation period.

As the complexity of ML models in irrigation management increases, the need for advanced computing resources becomes more pronounced. Many state-of-the-art models utilize GPUs or TPUs to significantly enhance performance by leveraging specialized hardware designed for parallel processing. This hardware acceleration is particularly crucial for complex deep learning models used in irrigation management, which often involve processing large amounts of spatial and temporal data. For instance, Bounoua et al. [98] compared the performance of convolutional long short-term memory (ConvLSTM) and convolutional neural network-long short-term memory (CNN-LSTM) models for predicting water stress in arboriculture using a time series of remote sensing images. These GPU-accelerated models demonstrated high accuracy in forecasting water stress, with the CNN-LSTM model excelling for longer sequences of input data.

#### 4.3.1. Deploying ML Models

The deployment of ML models in irrigation management systems requires careful consideration of various technologies and techniques to ensure efficient, scalable, and reliable performance in real-world agricultural settings. Key deployment strategies, frameworks, and optimization methods, such as TensorFlow Serving, Apache MXNet Model Server, ONNX Runtime, model compression, hardware acceleration, and distributed training, can be used to transform ML models into operational components of irrigation decision support systems (Table 4). These technologies enable efficient and scalable inference, interoperability between diverse environments, improved resource utilization, and accelerated training of complex models.

The dynamic nature of agricultural environments poses challenges for the effective deployment and continuous improvement in ML models. Online learning algorithms, which enable the real-time update of models based on incoming data, are particularly valuable in irrigation management, where conditions can change rapidly due to weather patterns, soil moisture fluctuations, and crop growth stages. For example, Gelete and Yaseen [97] developed a hybrid model combining deep learning, nonlinear system identification, and ensemble tree intelligence algorithms for pan-evaporation estimation, demonstrating significant improvements in prediction accuracy compared to single models by incorporating principles of online learning.

Figure 5 illustrates the automated data processing pipeline in the cloud for irrigation management, showcasing the seamless translation of raw sensor data into actionable irrigation insights and control commands. The process begins with the acquisition of diverse data types from IoT sensors and external sources, followed by preprocessing steps to ensure data quality and compatibility with ML models. The clean, processed data are then fed into deployed ML models for predictive modeling tasks, such as yield prediction, anomaly detection, and irrigation scheduling optimization. The generated insights are combined with constraint optimization techniques to create precise irrigation schedules and control parameters, which are translated into actuation commands for IoT-enabled irrigation systems.

The integration of IoT sensor data with cloud-based data sources is crucial for enhancing the accuracy and resolution of irrigation insights. Data fusion techniques, such as Kalman filtering, play a critical role in combining heterogeneous data sources to generate accurate and comprehensive irrigation recommendations by integrating real-time sensor data with broader spatial and temporal information [105]. Advanced machine learning approaches, like transfer learning with deep neural networks, further improve the accuracy of predictions for key variables like soil moisture content [106].

Various machine learning approaches, each with unique strengths and limitations, have been successfully applied for generating irrigation insights. Fuzzy logic, for instance, handles uncertainty and vagueness in decision-making using linguistic variables and rules, offering interpretability and the ability to incorporate expert knowledge [107]. Deep learning models, like LSTM networks and CNNs, excel at capturing complex spatiotemporal dependencies and enabling end-to-end learning, but typically require large, labeled datasets [108]. Reinforcement learning approaches learn optimal irrigation policies through trial-and-error interactions with the environment, adapting well to dynamic conditions but requiring careful design of state–action spaces and reward functions [109]. Regression models, such as PLSR and ANFIS, learn the relationships between input variables and irrigation decisions using statistical techniques, offering interpretability and ease of implementation but having limited ability to capture complex nonlinear relationships [110].

To close the loop between insights and action, ML-generated recommendations must be seamlessly translated into control commands for IoT-enabled irrigation systems. Protocols like MQTT and CoAP facilitate lightweight, real-time communication between sensors, actuators, and ML models [111,112]. Smart irrigation controllers exemplify this integration by automatically adjusting schedules based on weather data and soil moisture sensors to optimize water use [113].

The successful deployment and integration of ML models in irrigation management systems requires close collaboration between data scientists, agricultural experts, and IoT engineers. Key considerations include selecting appropriate model architectures and training techniques, ensuring data quality and consistency, optimizing model performance for resource-constrained edge devices, and designing intuitive user interfaces for monitoring and control.

More specialized approaches like ant colony optimization and model predictive control (MPC) have also shown promise in irrigation management. Ant colony optimization efficiently explores large search spaces and can handle discrete decision variables to find near-optimal solutions for irrigation schedules, although it may become stuck in local optima and requires tuning of algorithm parameters. MPC optimizes irrigation scheduling by solving finite-horizon optimization problems at each sampling time, considering constraints on state and control variables. It has demonstrated improved water conservation and increased crop yields but requires accurate models of the soil–plant–atmosphere system and computational resources for real-time optimization.

The integration of advanced ML techniques, IoT technologies, and cloud computing in irrigation management has the potential to revolutionize agricultural water use by enabling data-driven, adaptive, and precise irrigation scheduling. However, realizing this potential requires addressing challenges such as data scarcity, model interpretability, and robustness to varying environmental conditions. The future research should focus on developing more efficient and transferable models, improving data fusion techniques, and exploring hybrid approaches that combine the strengths of different ML algorithms. Additionally, efforts should be made to democratize access to these technologies, ensuring that smallholder farmers and resource-constrained regions can benefit from these advancements.

By leveraging the power of ML and seamlessly integrating it into the automated irrigation pipeline, new levels of precision, efficiency, and sustainability in agricultural water management can be unlocked. As these techniques are refined and advanced, the vision of fully autonomous, intelligent irrigation systems that dynamically optimize water allocation based on real-time insights comes closer to reality, promising a more resilient and productive future for agriculture in the face of growing water scarcity and climate change.

#### 4.3.2. Automated Application of Irrigation Insights Using Irrigation Systems

The integration of ML-generated insights with IoT-enabled irrigation control systems requires the adoption of appropriate architectures and protocols that facilitate lightweight, real-time communication [111,112]. The message queuing telemetry transport (MQTT) protocol and the constrained application protocol (CoAP) are well-suited for this purpose, offering efficient and reliable communication channels for exchanging data between sensors, actuators, and ML models [114]. Industry-leading products and services for smart irrigation demonstrate the practical application of these technologies. Smart irrigation controllers leverage weather data and soil moisture sensors to automatically adjust irrigation schedules based on real-time conditions, reducing water waste and promoting water conservation [113]. The message queuing telemetry transport (MQTT) protocol and the constrained application protocol (CoAP) are wellsuited for this purpose, offering efficient and reliable communication channels for exchanging data between sensors, actuators, and ML models [114].

Currently, the integration of IoT and machine learning technologies in automated irrigation systems has been applied to various irrigation methods. Center pivot systems, in particular, have evolved significantly beyond basic sprinkler capabilities. Modern pivot systems commonly incorporate multiple advanced technologies: corner arm systems that extend to irrigate field corners typically missed by standard circular patterns; GPS guidance for precise positioning and navigation; variable rate irrigation capabilities that allow different application rates across the field; and individual sprinkler controls for precise water distribution. While drip irrigation systems have seen extensive IoT integration for soil moisture monitoring and automated control [115], utilizing real-time sensor data and predictive models to optimize irrigation scheduling, the extent of adoption and comparative effectiveness across different system types remains an area of ongoing research. As research and development progress, the combination of IoT and machine learning is expected to advance automated irrigation across various system types, enabling more precise and adaptive water management strategies.

Different irrigation systems may have varying potential for automation due to factors such as precision, adaptability, and compatibility with advanced sensing and control technologies. Srivastava et al. [116] demonstrated that smart sprinkler systems can reduce water loss by up to 95% compared to conventional methods, while Jain et al. [117] showed automated drip systems effectively optimize soil moisture conditions. However, comprehensive comparative studies across all irrigation types specifically regarding their automation potential are limited, making it difficult to definitively rank their automation potential. Figure 6 presents an overview of various irrigation technologies, including sprinkler, drip, furrow, and flood irrigation, in terms of their current automation implementations and challenges.

Furrow irrigation has shown potential for automation through IoT sensor networks and automated flow control, as evidenced by Uddin et al.’s [121] successful implementation of a smart automated furrow system for cotton. Flood irrigation faces certain challenges for full automation, with limited research on successful IoT or ML applications. However, Pramanik et al. [122] achieved 86.6% irrigation efficiency improvement through an automated soil moisturebased system for basin layout. The implementation of fully automated control varies across these technologies, with each system type presenting unique opportunities and challenges based on their design and water application methods.

It is important to note that systems with higher levels of automation also face challenges such as higher initial costs, maintenance requirements, and energy consumption compared to traditional surface irrigation methods. Furrow and flood irrigation, while simple and low-cost, have traditionally had fewer implementations of automation and precision management technologies, which can affect water use efficiency and potential environmental impacts. However, the recent research has demonstrated progress in integrating IoT and ML technologies to enhance automation in various irrigation systems. For instance, Rajasekhar et al. [123] developed a fog-based intelligent irrigation system for furrow irrigation that utilizes LoRaWAN for data transmission and machine learning for rainfall prediction, demonstrating potential water savings of approximately 60%.

The environmental impacts of flood irrigation pose significant challenges globally, driving regulatory responses and affecting technology adoption patterns. Studies have shown that flood irrigation can lead to soil salinization, reduced water infiltration, and degradation of groundwater quality through salt loading and nutrient leaching [124]. These environmental concerns have prompted increasing regulatory restrictions on flood irrigation in many regions. For example, in Brazil, environmental agencies are implementing strict controls on flood irrigation due to its low efficiency and excessive water consumption [125].

Despite these challenges and the resulting regulatory pressures, research on the application of both IoT and ML technologies specifically to flood irrigation systems appears to be limited in the current literature. The relative difficulty of deploying fully automated control still varies across irrigation technologies, with sprinkler and drip systems generally being more amenable to automation than surface irrigation methods like furrow and flood irrigation. Key factors influencing the feasibility of automation include the complexity of the system design, the availability of compatible sensors and actuators, the variability in water distribution, and the potential for water and energy savings. Studies have shown that improved management practices and automation can significantly increase flood irrigation efficiency, with some systems achieving efficiency rates of up to 80% [126], suggesting potential for technological solutions to address both environmental and efficiency concerns. Further research is needed to explore the full potential of IoT and ML applications in automating both furrow and flood irrigation systems.

## 5. Interoperability, Standardization and Security

The vision of fully autonomous, scalable irrigation management hinges on the integration of diverse components within the system. This necessitates addressing interoperability, ensuring that sensors, actuators, controllers, and software platforms can communicate and exchange data effectively, regardless of their origins or underlying technologies [127]. Standardization complements this by providing a common framework for data formats, protocols, and interfaces, fostering consistency and compatibility across the entire irrigation management pipeline. The lack of widespread interoperability and standardization has presented a significant roadblock to the adoption of automated irrigation systems [15]. The prevalence of proprietary protocols and data formats creates difficulties in integrating components from various vendors into a unified system [128]. This fragmentation leads to elevated costs, increased complexity, and limited functionality. Table 5 summarizes the key data transfer and interoperability protocols used in irrigation systems, along with their key features and typical applications.

Interoperability in precision agriculture encompasses a range of protocols and standards, addressing everything from low-level sensor data transmission to high-level data exchange between farm management systems. The adoption of these technologies in real-world irrigation projects has demonstrated their potential to enable integration and unlock the benefits of precision agriculture.

Several key solutions have emerged to promote interoperability and standardization in this field.

At the data exchange level, agroXML offers a standardized format that facilitates communication between diverse software applications and platforms within the agricultural ecosystem [140]. This standardization allows automated irrigation systems to seamlessly share data with farm management software and other precision agriculture technologies [128]. Complementing this, ISOBUS (ISO 11783) provides a standardized communication protocol for agricultural machinery and implements. This protocol enables the integration of automated irrigation systems with other equipment, supporting advanced practices such as variable rate irrigation [24].

For efficient data transmission in IoT-based irrigation systems, the message queuing telemetry transport (MQTT) has become a popular choice [129]. This lightweight messaging protocol ensures reliable communication between devices and servers, crucial for real-time irrigation management [129]. Building on this, the OGC SensorThings API offers a RESTful interface for real-time sensor data access and management, facilitating seamless integration with web-based platforms and applications [22].

Finally, SensorML provides a standardized method for describing sensor characteristics and observations. This standardization potentially enables the easy integration of data from various sensor types into automated irrigation systems, further enhancing the interoperability of precision agriculture technologies [141].

### 5.1. Integration with Existing Irrigation Infrastructure

Transitioning from traditional irrigation methods to automated systems presents practical challenges, particularly in integrating the legacy infrastructure with modern technologies. One approach is to retrofit existing irrigation components with IoT sensors and control devices. Tan et al. [142] presented a smart irrigation system that enhances traditional time-based control methods by incorporating soil moisture and raindrop sensors connected to ESP32 and Arduino Uno microcontrollers. The system enables both manual and automatic irrigation modes through a user-friendly Thinger.io interface, demonstrating improved water conservation compared to conventional methods. This case study illustrates how legacy systems can be augmented with real-time data collection and wireless communication capabilities to optimize irrigation efficiency.

Another strategy involves developing low-cost frameworks that facilitate the integration of traditional irrigation practices with IoT-enabled water management solutions. Rana et al. [143] proposed a two-part system consisting of a watering system and a mobile app interface. In automatic mode, the water supply is connected to a pump that is activated based on soil moisture, humidity, and temperature data collected by sensors. The system has the potential to save 25% to 40% of water and energy resources compared to traditional irrigation methods. This example highlights how smart technologies can be designed to work alongside existing infrastructure, promoting compatibility and sustainable resource management.

### 5.2. Cybersecurity Considerations for Integrated Automated Irrigation Systems

The interconnected nature of IoT-based automated irrigation systems presents unique security challenges and vulnerabilities that require a comprehensive cybersecurity approach.

#### 5.2.1. Unique Security Risks and Vulnerabilities

These vulnerabilities are further amplified by the dynamic and distributed nature of agricultural environments, with limited connectivity, reliance on wireless communication, and the remote location of many farms. IoT-based automated irrigation systems are susceptible to various cybersecurity threats, including the following:Unauthorized Access and Data Breaches: remote access capabilities create potential entry points for unauthorized individuals, leading to manipulation of irrigation schedules, disruption of water flow, and unauthorized access to sensitive data [144].Data Tampering and Injection: malicious actors may tamper with or inject false data into the system, leading to sub-optimal irrigation practices and potentially harming crops or wasting water [144].Malware and Ransomware: the interconnectedness of devices increases the risk of malware and ransomware attacks, disrupting system functionality, compromising data integrity, and potentially crippling agricultural operations [145].Supply Chain Vulnerabilities: a compromise at any point in the complex supply chain of IoT-based irrigation systems can introduce vulnerabilities or backdoors exploitable by attackers [146].

#### 5.2.2. Best Practices and Strategies for Securing Automated Irrigation Systems

To mitigate these risks and enhance security, consider implementing the strategies outlined in Table 6. By adopting these best practices and strategies, automated irrigation systems can achieve a higher level of security and resilience, ensuring reliable and sustainable operation.

## 6. Monitoring and Ensuring System Reliability

Autonomous irrigation systems, promising as they are, must be built on a foundation of reliability and resilience. This means not only ensuring that the systems themselves function correctly, but also anticipating and mitigating potential points of failure, from faulty sensors to unpredictable environmental conditions—and even malicious intrusions. In this section, the strategies and techniques are explored for monitoring system status, building resilience against faults and failures, and implementing closed-loop control to adapt to changing conditions. Enhancing irrigation system resilience involves incorporating redundancy of critical components, such as sensors, controllers, and communication protocols [147]. For example, Sharma et al. [148] addressed the challenges of rural irrigation by proposing a photovoltaic (solar energy) system integrated with a Z-source inverter motor drive to power irrigation pumps. This setup includes a fault-tolerant circuit to handle switch failures, demonstrating the effectiveness of this redundancy strategy.

Implementing failover mechanisms, which automatically switch to backup components when primary systems fail, ensures a smooth transition and minimizes disruptions. Kaur and Bhattacharya [149] developed a green fault tolerant IoT-enabled mobile sink data collection scheme that provides fault-tolerant data collection in sensor networks through an AI-based rendezvous point selection and rotation mechanism. Simulations and testbed results validated the efficiency of this approach.

Future advances include AI-driven self-healing systems that can automatically identify and recover from faults [150]. Distributed architectures and edge computing contribute to resilience by decentralizing key functions and data flows, enabling localized control even during central failures [151,152]. For instance, Ribeiro and Kamienski [153] proposed a resilient fog-based IoT system for smart irrigation that maintains operation even during internet disconnections by storing and analyzing data locally at the edge. Their data value extraction mechanism reduced fog storage usage by 23–48% depending on crop growth stage.

Proactive approaches, such as unsupervised learning for anomaly detection and predictive maintenance models, can anticipate failures and enable targeted interventions to prevent downtime [154,155]. Benameur et al. [156] presented an innovative low-cost irrigation system employing deep learning to analyze anomalies in water usage, with autoencoders achieving 90–97% accuracy in detecting anomalies in soil moisture, air temperature, and humidity.

Figure 7 illustrates various fault tolerance techniques that work together to improve the reliability of precision irrigation systems. A truly robust automated irrigation system requires a multi-faceted approach encompassing physical redundancies (e.g., sensor duplication and communication channel backup), intelligent self-management (e.g., algorithmic diversity and adaptive control strategies), decentralized architectures (e.g., controller replication and edge computing), and predictive capabilities (e.g., anomaly detection and predictive maintenance using AI/ML). By integrating redundancy, diversity, reconfiguration, and graceful degradation strategies across the hardware, software, sensing, and control domains, irrigation systems can achieve high levels of fault tolerance and reliability even in the face of unexpected disruptions. Ongoing research into resilience evaluation methods will further enable quantitative assessment and optimization of these techniques for context-specific requirements [157].

A holistic reliability framework for automated irrigation should combine component-level hardening, system-level resilience architecture, and data-driven predictive analytics. This will enable self-awareness and self-adaptation in the face of both acute faults and gradual degradation, ultimately delivering robust and uninterrupted service over the full lifecycle of these mission-critical systems in agriculture.

## 7. Case Studies of Application of IoT-Based Automated Solutions in Irrigation Management Applications

To better understand the real-world implementation of autonomous irrigation, in this section, four case studies are analyzed, evaluating their effectiveness, cost, ease of use, and security considerations. These implementations demonstrate the practical application of the IoT architectures, devices, and protocols discussed in Section 3, showing how theoretical frameworks translate into functional irrigation systems. The case studies showcase the evolution of IoT-based irrigation from basic monitoring to sophisticated real-time control, providing concrete evidence of both capabilities and challenges in field deployments. Each case study is examined through the lens of the core components of an automated irrigation system, from data collection and processing to irrigation scheduling and application, highlighting both the benefits and limitations observed in practical deployments.

### 7.1. Case Study 1: Gutiérrez et al. [158]

The implementation by Gutiérrez et al. demonstrates the effectiveness of ZigBee-based wireless sensor networks in agricultural settings. Their approach prioritized system robustness and remote monitoring capabilities, as detailed in Table 7.

### 7.2. Case Study 2: Dursun and Ozden, [159]

Dursun and Ozden’s system architecture focused on reliability through redundancy, incorporating duplicate soil moisture sensors and fail-safe irrigation scheduling. Table 8 presents the key aspects of their implementation.

### 7.3. Case Study 3: Klein et al. [160]

Klein et al.’s implementation represents a significant advancement in precision irrigation, integrating multiple data sources for optimized water management. The system’s components and benefits are outlined in Table 9.

### 7.4. Case Study 4: Masseroni et al. [161]

The specialized system developed by Masseroni et al. addressed the unique challenges of rice cultivation, demonstrating the adaptability of autonomous irrigation technology to specific crop requirements. Table 10 summarizes their approach to automated water level management.

### 7.5. Key Takeaways from Case Studies

The four case studies presented in this review paint a vivid, albeit incomplete, picture of the current state of fully autonomous irrigation systems. They highlight the remarkable potential of these technologies to revolutionize agricultural water management, with examples like Gutíerrez et al. [158] and Dursun and Ozden [159] achieving impressive water savings of up to 90% through real-time data collection and closed-loop control. Klein et al. [160] even demonstrate a 26% yield increase using a satellite-guided variable rate drip irrigation system. These achievements offer a tantalizing glimpse into a future of optimized water use, enhanced crop productivity, and minimized manual labor.

Economic analyses from the case studies reveal a nuanced picture of cost benefit trade-offs across different scales of operation. Klein et al.’s sophisticated satellite-guided system, while highly effective, required a substantial investment of approximately USD 5000/hectare. This high initial cost highlights a critical challenge in scaling these technologies, particularly for smaller operations. However, recent research building on these case studies has demonstrated more accessible pathways to automation. For resource-constrained operations, innovative approaches combining predictive control with simplified hardware have shown potential to reduce system lifetime costs by 18–74% while improving irrigation reliability by 31–66% compared to conventional systems [162]. At medium scales, similar to those seen in the case studies, automated systems in intensive horticultural operations have achieved payback periods as short as 1.2 years through combined water savings and yield improvements [163]. Life-cycle analyses of different system types indicate that while initial setup costs range from EUR 3000–3500/ha, operational savings through reduced water and energy consumption can effectively offset these investments [164].

The significant upfront costs revealed in these case studies point to a critical need for enabling mechanisms to support adoption, particularly for smallholder farmers. While none of the case studies directly addressed financing solutions, subsequent implementations and policy research have identified several effective approaches. Government-supported credit guarantee schemes, particularly when structured through farmer groups establishing collective guarantee funds, have shown promise in reducing collateral requirements while maintaining borrower commitment through peer pressure [165]. These approaches work best when combined with specialized agricultural credit lines structured to match seasonal cash flows and simplified rural banking procedures. For example, in regions implementing systems similar to Gutíerrez et al.’s [158] design, farmer cooperatives have successfully leveraged collective purchasing power and shared maintenance responsibilities to make automation more accessible to individual members [166]. The success of these financing approaches ultimately depends on careful matching of technology sophistication to operational scale and available resources—a consideration notably absent from the presented case studies but crucial for wider adoption.

Even with financing support, the case studies and subsequent implementations highlight the need for adaptable and context-specific irrigation solutions that balance technological sophistication with economic viability. For instance, Dursun and Ozden’s system [159], while successful for dwarf cherry trees, might not translate effectively to larger, more diverse farms due to its lack of remote monitoring. Similarly, Masseroni et al. [161] found limited impact on water consumption and yield in rice paddies, suggesting the need for further refinement of automated systems for specific crop requirements and economic contexts. This pattern of varying success across different implementations reinforces the need to move away from a one-size-fits-all approach and invest in research and development that produces affordable solutions tailored to the diverse needs and resource constraints of farmers across varying contexts.

A critical lesson from examining transition costs across these cases is that the path to automation need not be all-or-nothing. While Klein et al.’s [160] comprehensive system demonstrates the full potential of automation, many farmers may benefit from a staged approach, beginning with basic monitoring and control infrastructure and gradually expanding capabilities as operational benefits are demonstrated. This staged implementation strategy, though not explicitly tested in the case studies, has emerged as a promising pathway for managing both technical complexity and financial constraints while building organizational capacity for more advanced automation [167,168].

Perhaps most concerning is the lack of focus on cybersecurity across all case studies. As we increasingly entrust critical infrastructure to data-driven automation, the potential consequences of cyberattacks, from disrupted water supply to manipulated sensor data, could have devastating effects on agricultural production and food security. While some studies briefly mention data security, none delve into the specific vulnerabilities and mitigation strategies essential for safeguarding these interconnected systems. This gap in the research leaves a critical question unanswered: how do we protect these increasingly vital systems from cyberattacks that could disrupt water supply, manipulate sensor data, and ultimately threaten food security?

The case studies, while demonstrating the transformative potential of autonomous irrigation, also expose crucial areas for further research and development. The highlighted challenges underscore the need for a more holistic approach in system design. Rather than focusing solely on individual performance metrics like water savings or yield increase, future research should prioritize economically viable solutions that integrate seamlessly with existing infrastructure, incorporate robust cybersecurity frameworks, and are adaptable to specific crop needs and local conditions. Only through such a comprehensive approach, encompassing both technological advancement and responsible implementation, can we unlock the full potential of autonomous irrigation for a more sustainable and equitable agricultural future.

## 8. Conclusions

In this review, a field brimming with potential is revealed, yet grappling with fundamental challenges as it strives towards truly autonomous irrigation. While the integration of IoT, machine learning, and cloud technologies promises optimized water use, increased yields, and reduced labor, the journey toward widespread adoption faces significant obstacles.

A key takeaway is the disconnect between technological capability and practical implementation. The case studies demonstrate impressive successes, achieving significant water savings and showcasing the power of data-driven, closed-loop control. However, these examples are often confined to specific contexts and struggle to scale effectively. The cost of upgrading legacy systems, particularly for smallholder farmers, remains a formidable barrier. Furthermore, the lack of standardized data formats and communication protocols hinders interoperability, creating a fragmented landscape of proprietary solutions that limit functionality and increase complexity.

Perhaps the most pressing concern is the vulnerability of these interconnected systems to cyberattacks. Ensuring robust cybersecurity measures becomes paramount as we entrust critical infrastructure to data-driven automation. The potential consequences of breaches extend far beyond economic losses, threatening food security and environmental sustainability.

The path forward requires a shift from isolated technological advancements to a more holistic and collaborative approach. Future research and development must prioritize human-centered design, ensuring that interfaces and decision-support tools are intuitive and tailored to farmers’ needs. Robustness and resilience must be built into system architectures, incorporating redundancy, fault-tolerance, and self-healing capabilities to guarantee reliable operation. Decentralized architectures leveraging edge computing and advanced AI can empower localized control, reducing reliance on continuous cloud connectivity. Critically, developing comprehensive cybersecurity frameworks is non-negotiable. Finally, embracing open standards and protocols is crucial for fostering interoperability, reducing vendor lock-in, and accelerating innovation.

Our analysis of implemented systems highlights several practical pathways for adoption. The most successful deployments typically begin with basic monitoring and control infrastructure, gradually expanding capabilities as operational benefits are demonstrated. Edge computing solutions have proven particularly valuable for areas with limited connectivity, while cloud-integrated systems offer the highest degree of automation for operations with robust infrastructure. The choice between these approaches should be guided by local conditions, existing infrastructure, and organizational capacity for technical support. This staged implementation strategy helps organizations build expertise while managing costs and complexity.

The transition to truly autonomous and intelligent irrigation demands a collaborative effort that transcends disciplinary boundaries. Researchers, technology developers, policymakers, and farmers must work together to address the technical, social, economic, and ethical dimensions of this transformative shift. By prioritizing human needs, fostering robust security, and promoting open collaboration, we can unlock the full potential of automated irrigation, paving the way for a more sustainable and equitable agricultural future.

## Figures and Tables

**Figure 1 sensors-24-07480-f001:**
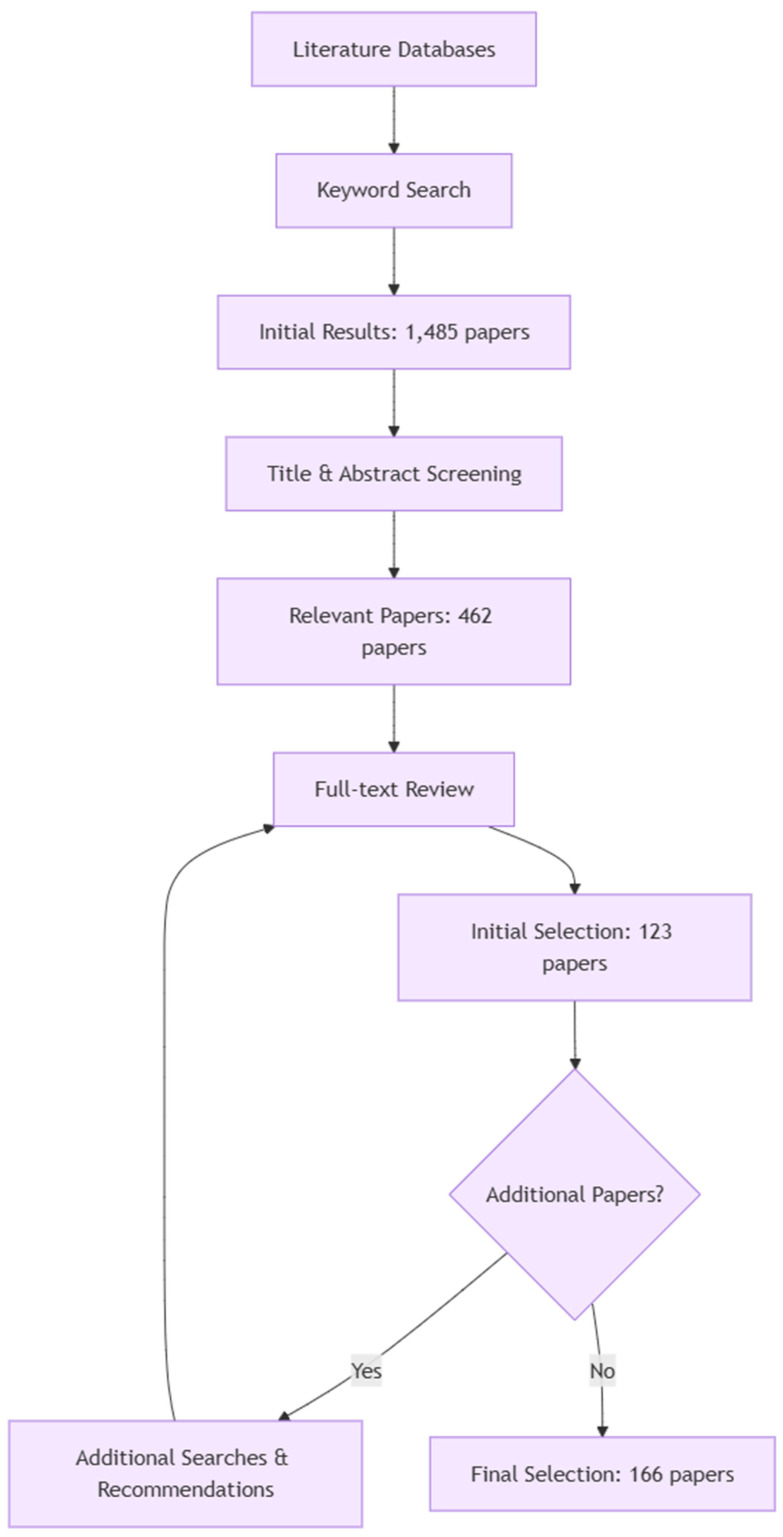
Flowchart depicting the literature search and selection process.

**Figure 2 sensors-24-07480-f002:**
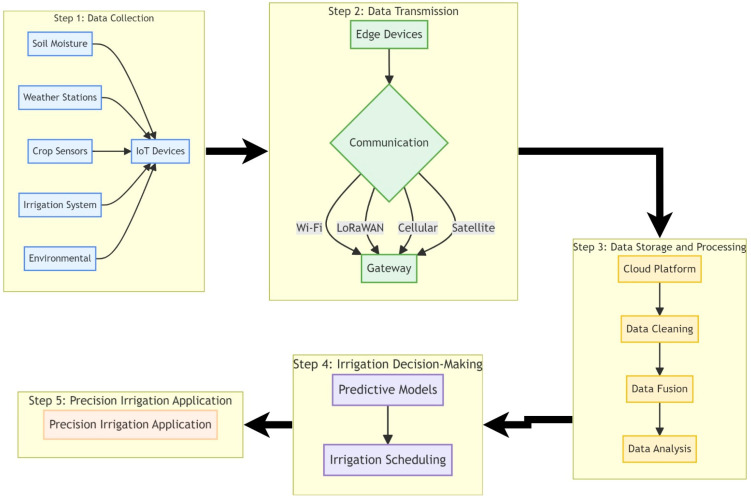
Automated irrigation system architecture depicting the flow from data collection through IoT devices, transmission via various networks, cloud processing, decision-making, to precision application.

**Figure 3 sensors-24-07480-f003:**
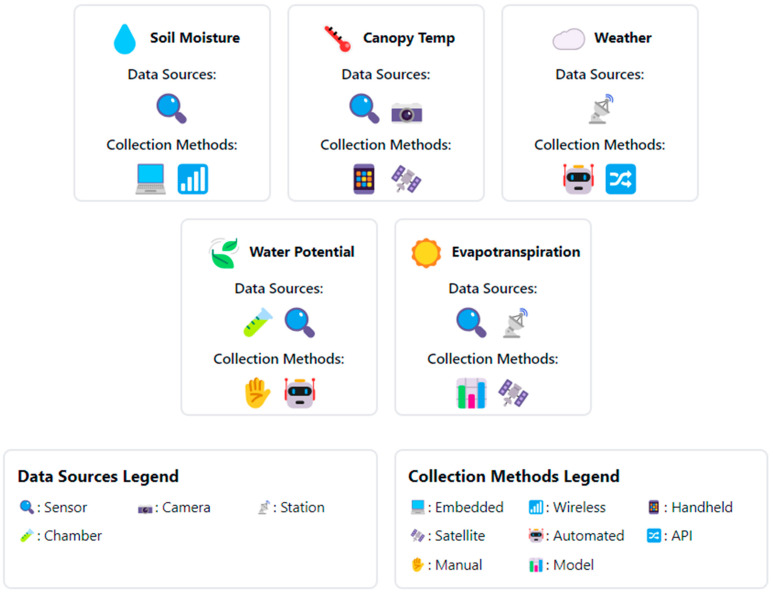
Common data types, sources, and collection methods in automated irrigation systems.

**Figure 4 sensors-24-07480-f004:**
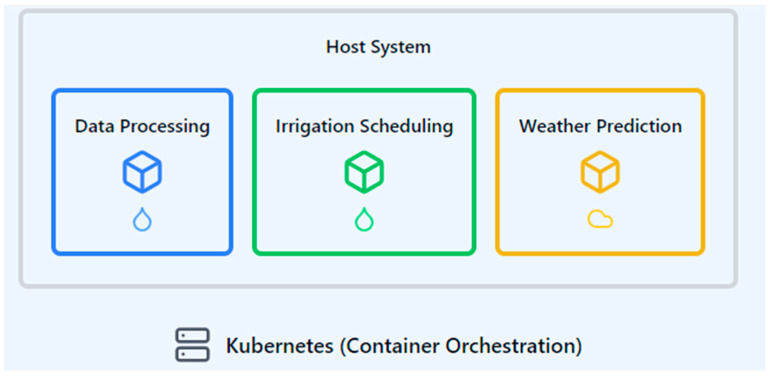
Containerized irrigation management system. Just as shipping containers standardize cargo transport, software containers can package irrigation system’s software components. Each container (color-coded) encapsulates a specific function, allowing for efficient deployment and scalability across the system, orchestrated by systems like Kubernetes.

**Figure 5 sensors-24-07480-f005:**
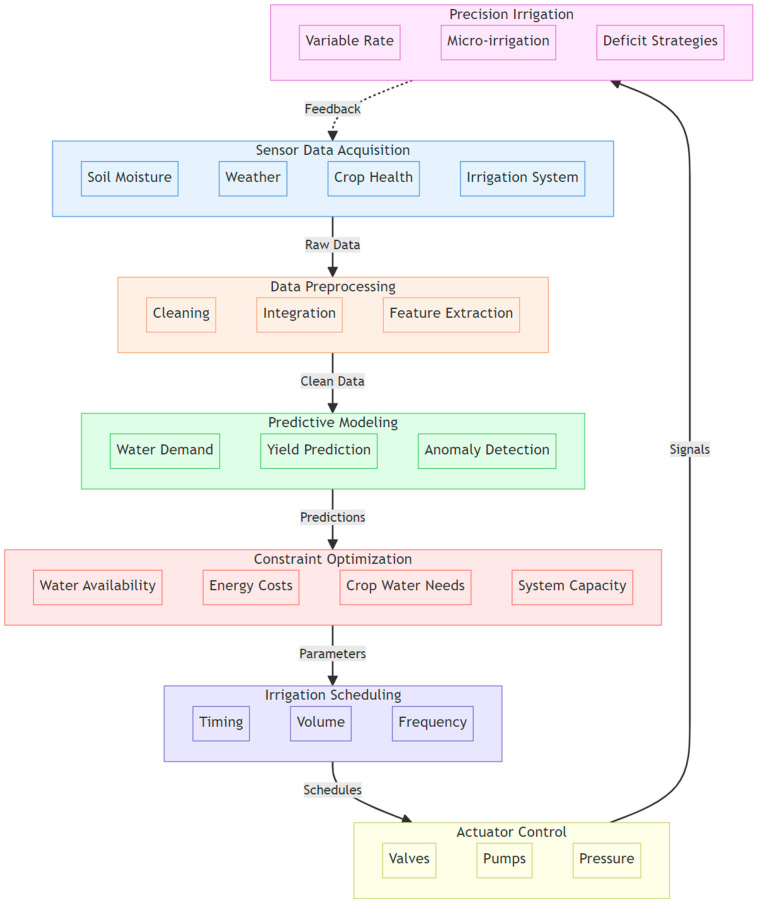
Automated data processing pipeline in the cloud for irrigation management.

**Figure 6 sensors-24-07480-f006:**
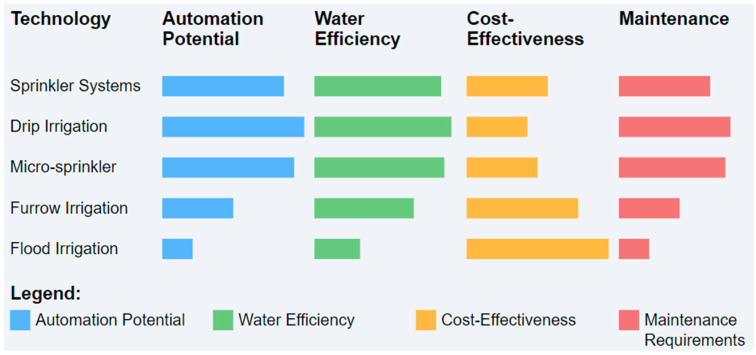
Comparison of irrigation technologies across four key metrics. Longer bars indicate better performance (higher potential/efficiency, higher cost-effectiveness, lower maintenance). Data synthesized from Musick et al. [118], C’orcoles et al. [119], and Evans [120]. Specific values may vary based on implementation and local conditions.

**Figure 7 sensors-24-07480-f007:**
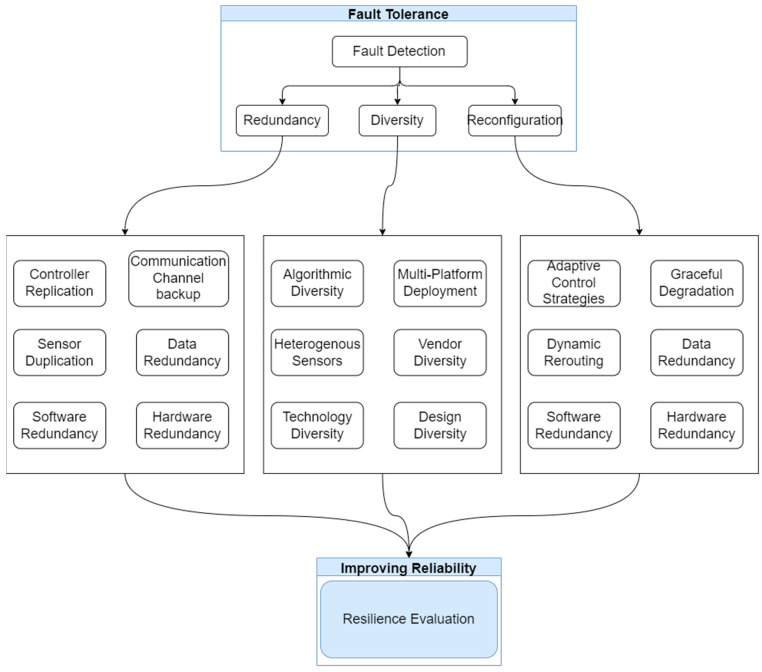
Fault tolerance techniques for improving reliability in precision irrigation systems.

**Table 1 sensors-24-07480-t001:** Research questions and hypotheses framework for analyzing automated irrigation systems.

Objective	Specific Objective	Research Question	Motivation	Hypothesis
Examine the Automation of the Irrigation Management Pipeline	Data Collection to Cloud	What IoT devices and communication protocols are most effective for real-time data collection and transmission in diverse agricultural environments?	To understand the current state-of-the-art in data acquisition and transmission technologies for irrigation management.	Low-power wide-area networks (LPWANs) and edge computing are increasingly prevalent for efficient and reliable data collection and transmission in automated irrigation systems.
		How are data quality and preprocessing techniques ensuring accurate and reliable data for decision-making in automated irrigation systems?	To assess the effectiveness of data quality control and preprocessing methods in ensuring reliable data for irrigation management.	Advanced data filtering, outlier detection, and imputation techniques significantly improve the accuracy and reliability of data used for irrigation decision-making.
	Data Processing and Decision Making in the Cloud	What machine learning algorithms and cloud-based platforms are being utilized for predictive modeling and irrigation scheduling in automated systems?	To investigate the current trends and advances in machine learning and cloud computing for irrigation management.	Deep learning models and cloud-based platforms are increasingly adopted for accurate and adaptive irrigation scheduling in automated systems.
		How are online learning algorithms and real-time data integration enabling adaptive and responsive irrigation management?	To explore the role of online learning and real-time data integration in achieving dynamic irrigation control.	Online learning algorithms enable automated irrigation systems to continuously adapt to changing conditions and optimize water use efficiency.
	Irrigation Scheduling and Application	How are ML-generated insights being translated into actionable irrigation commands and applied to various irrigation systems (e.g., sprinkler, drip, surface)?	To examine the effectiveness of automated systems in controlling different types of irrigation infrastructure.	Automated irrigation systems utilizing ML-generated insights can effectively control various irrigation technologies, optimizing water delivery and minimizing waste.
		What closed-loop control techniques are being employed to ensure precise and adaptive irrigation based on real-time feedback?	To understand the role of closed-loop control in achieving dynamic and responsive irrigation management.	Closed-loop control techniques, such as PID controllers and model predictive control, are crucial for achieving precise and adaptive irrigation based on real-time feedback.
Analyze the Effectiveness and Efficiency of Integrated End-to-End Automated Irrigation Systems		What are the documented benefits and limitations of fully autonomous irrigation systems in terms of water use efficiency, crop yield, and labor productivity?	To assess the overall impact of automated irrigation systems on agricultural practices and resource management.	Fully autonomous irrigation systems can significantly improve water use efficiency, crop yield, and labor productivity compared to traditional irrigation methods.
Investigate the Role of Interoperability and Standardization		What are the key challenges and opportunities for achieving interoperability and standardization in automated irrigation systems?	To identify the barriers and potential solutions for integration of diverse components within automated irrigation systems.	The lack of standardized data formats and communication protocols hinders the interoperability of automated irrigation systems.
Identify Gaps and Propose Solutions for Seamless Integration		What research gaps and future directions need to be addressed to achieve fully autonomous and scalable irrigation management?	To outline the future research needs and potential advancements in the field of automated irrigation management.	Future research should focus on developing more robust, scalable, and cost-effective solutions for automated irrigation management, addressing challenges related to data quality, security, and interoperability.

**Table 2 sensors-24-07480-t002:** Key techniques employed for data cleaning, preprocessing, and fusion.

Technique	Description	Benefits	Citations
Filtering	Recursively estimates the system’s state based on previous measurements and current sensor data, considering noise and uncertainty. Effective for handling missing values and correcting erroneous readings.	Improves data accuracy and reliability by mitigating the impact of noise and sensor malfunctions.	[69]
Moving Average	Averages consecutive data points to provide a more stable representation of the underlying trend, filtering out short-term fluctuations.	Smooths out data and reduces the impact of temporary variations, leading to a more accurate representation of trends.	[70]
Adaptive Thresholding	Dynamically adjusts thresholds based on the statistical properties of the data to effectively identify anomalies and minimize false positives.	Accurately identifies outliers while minimizing the risk of incorrectly flagging normal data points as anomalies.	[71]
Data Normalization	Transforms data values to a common scale (e.g., min–max scaling, z-score normalization) to ensure all features contribute equally to the analysis.	Prevents features with larger values from dominating the analysis and ensures all features are considered equally.	[72]
Feature Scaling	Optimizes the range of feature values to improve the performance and convergence of machine learning models (e.g., standardization, normalization).	Reduces the influence of outliers and enhances the model’s ability to learn from the data, leading to improved accuracy and performance.	[73]
Data Fusion	Integrates information from diverse sources to create a more comprehensive and reliable dataset.	Provides a holistic understanding of crop water requirements and environmental conditions by combining data from various sensors and sources, leading to more informed and accurate irrigation decisions.	[74]

**Table 3 sensors-24-07480-t003:** Key technologies and their benefits for containerization.

Technology	Description	Benefits	Citations
Docker	Package applications and their dependencies into self-contained units (containers) for consistent and reproducible execution across different platforms.	Enables efficient resource utilization, isolation, and portability of applications.	[89]
Kubernetes	Manages the deployment, scaling, and networking of containers across a cluster of machines.	Facilitates scalability, automation, and efficient resource management for containerized applications.	[90]
Auto-scaling	Automatically adjusts the number of container instances based on real-time demand.	Ensures sufficient resources are available during peak workloads while avoiding over-provisioning during low demand periods.	[91]
Dynamic Resource Allocation	Fine-grained adjustment of resources allocated to each container based on its specific needs and the current workload.	Optimizes resource utilization and ensures each container has the necessary resources to perform its tasks effectively.	[92]

**Table 4 sensors-24-07480-t004:** Technologies, frameworks, and optimization techniques for deploying ML.

Framework	Description	Benefits	Citations
Serving	High-performance system for serving TensorFlow models.	Efficient and scalable inference, suitable for real-time applications with low latency and high throughput requirements.	[99]
Apache MXNet Model Server	Flexible and efficient solution for deploying models trained with MXNet, supporting a wide range of deep learning models and inference backends.	Versatility and support for various deep learning models, suitable for complex irrigation systems utilizing different types of ML models.	[100]
ONNX Runtime	Cross-platform inference engine compatible with various ML frameworks, including PyTorch, TensorFlow, and MXNet.	Enables deployment of models in diverse environments, facilitating interoperability and reducing the need for model conversion.	[101]
Model Compression	Techniques such as pruning and quantization to reduce the size and computational requirements of ML models without compromising accuracy.	Improves model efficiency and reduces resource consumption, making it suitable for resource-constrained environments or real-time applications.	[102]
Hardware Acceleration	Utilizes GPUs or TPUs to significantly enhance model performance by leveraging specialized hardware designed for parallel processing.	Reduces inference time and enables real-time processing of sensor data for timely irrigation decisions.	[103]
Distributed Training	Techniques like Horovod and BytePS to distribute training across multiple machines, parallelizing the process and reducing training time.	Enables training of complex models with large datasets in a shorter time, improving model accuracy and efficiency.	[104]

**Table 5 sensors-24-07480-t005:** Key data transfer and interoperability protocols used in irrigation systems.

Protocol	Key Features	Relevant Standards	Typical Applications	Example Implementations
MQTT	Lightweight, publish-subscribe, suitable for low-bandwidth networks	ISO/IEC 20922 [129]	Sensor data transmission, device control, data integration	[130,131]
CoAP	RESTful, low overhead, designed for resource-constrained devices	RFC 7252 [80]	Device management, data exchange, service discovery	[117,132]
OGC SensorThings API	RESTful, geospatial-enabled, facilitates IoT integration	OGC 15-078r6 [22]	Sensor data management, geospatial analysis, data visualization	[133,134]
AgGateway ADAPT	Open-source, agriculture-specific, enables plug-and-play compatibility	ANSI/ASABE S632 [135]	Farm management information systems, precision agriculture data exchange	[136]
agroXML	XML-based, facilitates data exchange between agricultural software	ISO 11783 [23]	Farm management, precision agriculture, traceability	[137]
FIWARE	Platform-agnostic, facilitates app development and integration	FIWARE [138]	Smart farming, food supply chain management, collaborative business networks	[139]

**Table 6 sensors-24-07480-t006:** Best practices and strategies for securing automated irrigation systems.

Strategy	Description	Citations
Secure Device Provisioning and Authentication	Implement strong authentication mechanisms, such as multi-factor authentication and hardware security modules, to prevent unauthorized access	[144]
Encryption and Secure Communication Protocols	Employ robust encryption algorithms and secure communication protocols to protect data confidentiality and integrity	[145]
Regular Firmware and Software Updates	Maintain a regular update schedule for firmware and software to address vulnerabilities and mitigate emerging threats	[144]
Network Segmentation and Access Control	Divide the network into smaller, isolated sub-networks and implement access control mechanisms to limit the impact of a security breach	[146]
Intrusion Detection and Prevention Systems	Implement intrusion detection and prevention systems (IDPS) to monitor network traffic and system activity, identifying and blocking suspicious behavior	[144]
Security Audits and Penetration Testing	Regularly conduct security audits and penetration testing to identify vulnerabilities before they can be exploited	[144]
User Awareness and Training	Educate farmers and irrigation managers about cybersecurity best practices to minimize human error and promote security awareness	
Incident Response Planning	Establish a comprehensive incident response plan to effectively handle security incidents and minimize their impact	

**Table 7 sensors-24-07480-t007:** Key aspects of the fully autonomous irrigation.

Aspect	Details *
Step 1: Real-time data collection and transmission	Hourly transmission of sensor data from WSUs to WIU via ZigBee; remote access to system status and performance data through GPRS uplink
Step 2: Data storage and processing	Edge computing using microcontrollers; sensor data uploaded to web server for remote monitoring
Step 3: Predictive modeling and irrigation scheduling	Soil moisture and temperature threshold-based control algorithm; fixed irrigation durations triggered when thresholds exceeded
Step 4: Application of insights	Solenoid valves directly controlled by digital outputs of WIU microcontroller; irrigation amounts based on real-time sensor feedback
System security, robustness, and monitoring	Alert notifications sent via GPRS for system failures; solar power and rechargeable batteries for continuous operation; web application enables real-time remote monitoring
Cost and ease of use	Estimated total cost of USD 1380 for electronic components; designed for low cost and use in geographically isolated areas; modular design allows easy system expansion
Benefits and limitations	Water savings of up to 90% compared to traditional irrigation practices; solar-powered, enabling use in remote areas without grid power; remote alerts help quickly identify and resolve issues

* WSU: wireless sensor unit; WIU: wireless information unit; GPRS: general packet radio service.

**Table 8 sensors-24-07480-t008:** Key aspects of the fully autonomous irrigation system.

Aspect	Details *
Step 1: Real-time data collection and transmission	Soil moisture data sent from SU to BSU every 2 min via 434 MHz RF; two-way wireless communication between base station, valve, and sensor units
Step 2: Data storage and processing	Sensor data processed by PIC microcontrollers in BSU, VU, and SU; BSU logs all sensor data and valve status on SD card for local monitoring
Step 3: Predictive modeling and irrigation scheduling	Soil moisture threshold-based control; BSU evaluates soil moisture data and sends control signals to VU for valve actuation
Step 4: Application of insights	Solenoid valves controlled by VU based on signals from BSU; real-time soil moisture feedback used to trigger irrigations
System security, robustness, and monitoring	BSU switches to scheduled irrigation if no data received from SU; duplicate soil moisture sensors at each SU for redundancy; all units solar-powered for uninterrupted off-grid operation
Cost and ease of use	Low component costs enable cost-effective implementation; designed for cost effectiveness and use by growers; data logging enables performance analysis and irrigation optimization over time
Benefits and limitations	Reduces time spent by workers on irrigation management; maintains optimal soil moisture levels in root zone of trees; no significant yield increase after 1 season; lack of remote monitoring and control capabilities

* SU: sensor unit; BSU: base station unit; RF: radio frequency; PIC: peripheral interface controller; VU: valve unit.

**Table 9 sensors-24-07480-t009:** Key aspects of the fully autonomous irrigation.

Aspect	Details *
Step 1: Real-time data collection and transmission	Control nodes report valve status and flow meter data hourly via wired RS485 bus; cellular link between central computer and cloud platform
Step 2: Data storage and processing	Irrigation analytics and scheduling performed in cloud and transmitted to central computer; cloud platform aggregates and cleans flow meter and valve data for analysis
Step 3: Predictive modeling and irrigation scheduling	Water delivery optimized based on yield and water efficiency objectives; irrigation amounts calculated from satellite NDVI data and reference evapotranspiration; management zones delineated based on historical yield data and NDVI
Step 4: Application of insights	Solenoid valves in control nodes directly controlled by central computer commands; weekly irrigation scheduling adapted based on latest satellite imagery
System security, robustness, and monitoring	Redundant power and communication lines between control nodes; solar-powered control nodes and backup batteries for continuous operation
Cost and ease of use	Costs about USD 5000/hectare, with an estimated 2.5 years ROI for wine grapes; 1.6–2.5 year payback estimated for various high-value crops; optimum control resolution of 30 m to balance costs and benefits
Benefits and limitations	% average increase in water use efficiency and 26% yield increase over 2 years; significantly improved yield uniformity; fully automated system with minimal human input; high upfront costs and technical complexity as barriers to adoption

* RS485: A standard defining the electrical characteristics of drivers and receivers for use in serial communications systems; NDVI: normalized difference vegetation index; ROI: return on investment.

**Table 10 sensors-24-07480-t010:** Key aspects of the fully autonomous irrigation system.

Aspect	Details
Step 1: Real-time data collection and transmission	Water level and flow data collected every 10 min; sensor data wirelessly transmitted to Gateway, uploaded to cloud via cellular
Step 2: Data storage and processing	Local control by FarmConnect Gateway; cloud-based software for data visualization and gate control parameter configuration; FarmConnect Gateway filters and aggregates raw sensor data before cloud upload
Step 3: Predictive modeling and irrigation scheduling	Maintain target water levels in paddies by automatically actuating gates; performance assessed in water use efficiency and maintenance of constant flooding
Step 4: Application of insights	BayDrive gates automatically control flow based on water level feedback from sensors; real-time water level monitoring used for closed-loop control of gates
System security, robustness, and monitoring	Sensors and gates connected to Gateway for centralized status monitoring; failure prevention measures implemented, but SMS alerts not explicitly mentioned; powered sensors and Gateway for reliable operation, but fully off-grid capability not specified in the paper
Cost and ease of use	Cost of EUR 638–689 per hectare suitable for farmers; software and gate control settings need to be tailored to traditional rice irrigation practices; conduct training to help farmers select appropriate target water levels
Benefits and limitations	Drastically reduces time spent by workers controlling gates and water levels; automatic system successfully maintains stable water levels; no significant reduction in water consumption or yield increase; growers tend to set higher than necessary target water levels

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
