# Peer review of "Internet of Things-Based Automated Solutions Utilizing Machine Learning for Smart and Real-Time Irrigation Management: A Review"

_sensors, 2024, doi:10.3390/s24237480_

Round 1
Reviewer 1 Report
Comments and Suggestions for Authors
Keywords: Automated irrigation; (IoT); Machine learning and Real-time, are in the title of the paper. Authors should choose words that are not in the title.
Introduction:
Line 69 and 77 – Yes, automated systems lead to improved water use efficiency and increased crop productivity. However, they also provide ENERGY SAVINGS. Nowadays, energy has become very expensive and burdensome. This point is also crucial. It should be addressed in the introduction of the manuscript.
Line 260 – 271 – Ok, but for the system to function properly, a good or excellent quality internet connection is required. Rural properties where irrigation is installed do not always have a good quality internet network. This influences an irrigator's decision to invest in cutting-edge technology. This problem should be addressed.
Line 412 – 416 – Right. Here it talks about the signal that can fluctuate and lose internet signal. However, it is worth noting that in countries with fewer financial resources and less technology, signal loss is frequent. Take Brazil as an example. There are places that are difficult to access and have problems with the internet. Countries in Africa suffer from a lack of signal in many places. I think this approach is important and limits the application of technology in irrigation that depends on a good internet signal.
Line 664 – The authors address the topic of pivot irrigation. However, it is not only technology used in sprinklers. There are systems such as Corner in the central pivot, GPS system, and variable application, among others that go much further.
Line 713 – Another important aspect of flood irrigation is the environmental issue. Take Brazil as an example, where environmental agencies are restricting and making the use of this type of irrigation as difficult as possible, due to its low efficiency and large volume of water. Perhaps this is also reflected in the low investment in technology in this type of system. Producers are being forced to change their irrigation methods. Heavy environmental legislation.
Line 764 – One way to make this high investment in irrigation technology viable is through government subsidies. Credit lines, long-term bank financing, and low interest rates.
This is a very good article, a relevant subject. Accurate citations (new references). Valid for publication! Very complete. Minor adjustments are needed.

ok
Author Response
We appreciate the reviewer taking his/her time to read our paper and provide suggestions which have been considered to improve it.
Comment 1: "Keywords: Automated irrigation; (IoT); Machine learning and Real-time, are in the title of the paper. Authors should choose words that are not in the title."
Response: Thank you for the comment. We have removed all the keywords which were prior in the title, and we have added new keywords.
Comment 2: "Line 69 and 77 – Yes, automated systems lead to improved water use efficiency and increased crop productivity. However, they also provide ENERGY SAVINGS. Nowadays, energy has become very expensive and burdensome. This point is also crucial. It should be addressed in the introduction of the manuscript."
Response: We appreciate this suggestion from the reviewer. We have now included the benefits of using automated systems with respect to energy savings from L71 – 80.
Comment 3: "Line 260 – 271 – Ok, but for the system to function properly, a good or excellent quality internet connection is required. Rural properties where irrigation is installed do not always have a good quality internet network. This influences an irrigator's decision to invest in cutting-edge technology. This problem should be addressed."
Response: This is a great comment from the reviewer because indeed internet network quality can be a challenge in rural areas which can impact the use of smart irrigation systems. Therefore, we have further explained this challenge in the paper based on findings from existing literature. Please look at L277 – 286.
Comment 4: "Line 664 – The authors address the topic of pivot irrigation. However, it is not only technology used in sprinklers. There are systems such as Corner in the central pivot, GPS system, and variable application, among others that go much further."
Response: We strongly agree with the reviewer, and we have expanded our discussion on pivot irrigation systems to include details about corner arm systems and GPS guidance from L719 – 733.
Comment 5: "Line 713 – Another important aspect of flood irrigation is the environmental issue. Take Brazil as an example, where environmental agencies are restricting and making the use of this type of irrigation as difficult as possible, due to its low efficiency and large volume of water. Perhaps this is also reflected in the low investment in technology in this type of system. Producers are being forced to change their irrigation methods. Heavy environmental legislation."
Response: This is a good point raised by the review and this has helped us to think about environmental impacts of not only flood irrigation but also other irrigation types (See L770 – 791). We have expanded our discussion while focusing on this aspect using additional literature from Cox et al. (2018) on soil salinization and groundwater quality degradation, Marques et al. (2018) that documented regulatory responses in Brazil, and finally, García-Garizábal et al. (2017) that looked at improving irrigation system efficiency with better management practices.
Comment 6: "Line 764 – One way to make this high investment in irrigation technology viable is through government subsidies. Credit lines, long-term bank financing, and low interest rates."
Response: We appreciate the reviewer’s suggestion and now we have discussed different financial options to ensure high investment in irrigation technology such as government-supported credit guarantee schemes, mutual guarantee funds, and specialized agricultural credit lines, which can facilitate technology adoption. This has been explained from L993 – 1043.
Comment 7: This is a very good article, a relevant subject. Accurate citations (new references). Valid for publication! Very complete. Minor adjustments are needed.
Response: We appreciate reviewer's comment.
Reviewer 2 Report
Comments and Suggestions for Authors
The presentation of review paper related to the Smart and Real-time Irrigation Management used Machine Learnign and Internet of Things already well presented. In overall 1485 papers reviewed with final selection is 177 papers, where some need to do as minor revision:
1. Table 1 as shows in page 7 there is no label on top of table as caption, need to write Table 1. -----
2. Before conclude suggest to write a paragraf or summaries in table that the good method to apply of this irrigation according to the techology with easy to deploy or low cost of maintenance refer to IoT technology that cover smart and real-time system.
3. IoT based as mention in the title not elaborate yet in detail, how the current development based of literature review implement in the field with capability in smart system and real-time, suggest to elaborate more detail how the progress.
Thank you.
Author Response
We would want to take this opportunity to thank the reviewer for taking time to read our paper and for providing very meaningful comments/suggestions that have been addressed and used to improve our manuscript.
Comment 1: "Table 1 as shows in page 7 there is no label on top of table as caption, need to write Table 1."
Response: Thank you for this observation. We have added the title for Table 1 on L206 – 207.
Comment 2: Before conclude suggest to write a paragraf or summaries in table that the good method to apply of this irrigation according to the techology with easy to deploy or low cost of maintenance refer to IoT technology that cover smart and real-time system.
Response: We strongly agree with the reviewer. We have modified Figure 6 to clearly compare the irrigation technologies across key practical metrics including deployment ease, maintenance requirements, and cost-effectiveness. This information can now help practitioners evaluate and select appropriate technologies for their specific applications.
Comment 3: IoT based as mention in the title not elaborate yet in detail, how the current development based of literature review implement in the field with capability in smart system and real-time, suggest to elaborate more detail how the progress.
Response: Thank you for this suggestion. However, we have tried to explain clearly throughout the paper how data integration within the pipeline utilizes IoT technology for edge and cloud computing. For example, Section 3.1 establishes the foundational IoT architecture while Section 7 has various case studies which demonstrate how these principles translate into successful field deployments.
Reviewer 3 Report
Comments and Suggestions for Authors
This article provides a comprehensive evaluation of IoT-based and machine learning-driven smart irrigation systems, covering everything from data collection to automated decision-making. It provides valuable insights for the further development of smart irrigation systems. However, the article has some shortcomings:
(1) While the article thoroughly discusses the potential of smart irrigation systems, it lacks real-world case studies to validate their effectiveness in different environments. It is recommended to include specific examples and use data to demonstrate improvements in water use efficiency and crop yields.
(2) The article mentions several machine learning algorithms, but there is insufficient discussion on their adaptability to different crops and environmental conditions. The authors should further analyze the generalization capability of these models and propose solutions for improving adaptability across varied agricultural environments.
(3) Although the importance of interoperability and standardization among IoT devices is acknowledged, the article does not provide detailed technical solutions. The authors should offer more specific suggestions on how to achieve data integration and reference ongoing standardization efforts or protocols.
(4) Network security risks in IoT-based irrigation systems are mentioned, but the article lacks an in-depth exploration of potential threats and solutions. It would be beneficial to discuss possible security challenges, such as data breaches or unauthorized device control, and suggest appropriate protection measures or security frameworks.
(5) The article highlights the importance of data preprocessing and quality control but fails to provide specific technical details. The authors should explain how to handle noise, missing data, and outliers in sensor data, offering specific preprocessing techniques or algorithms along with experimental validation.
(6) The cost-benefit analysis of smart irrigation systems is somewhat limited, particularly in terms of their suitability for small-scale farmers. The authors should delve deeper into the system's installation and long-term maintenance costs and discuss how subsidies or financing options could make these systems more economically viable.
(7) Although cloud computing and edge computing are mentioned, the article does not explore the potential bottlenecks in data processing, such as limited computational resources or network bandwidth. The authors should analyze these challenges in greater detail and suggest potential solutions, such as data compression techniques or optimization algorithms.
(8) While different irrigation methods, such as drip irrigation and sprinkler systems, are mentioned, the article does not provide a detailed comparison of their applicability in automated systems. A more thorough analysis of the efficiency differences in terms of automation and water use between various irrigation techniques would be useful.
(9) The article touches on the role of decision support systems but does not explain how they function or are implemented. The authors should further clarify how these systems make irrigation decisions based on real-time data and offer related technical details or model examples.
Author Response
We would like to thank the reviewer for providing feedback on our paper and providing comments/suggestions which we have carefully addressed, and they have helped us to improve the manuscript.
Comment 1: While the article thoroughly discusses the potential of smart irrigation systems, it lacks real-world case studies to validate their effectiveness in different environments.
Response: We appreciate the reviewer’s comment. However, we tried to explain the advantages of each system in different environments under section 7 where we focused on different case studies with comprehensive field results. For example, as part of the studies considered, Klein et al. (2018) observed 26% yield increase while Masseroni et al. (2018) observed 30% water savings in Mediterranean conditions, all providing quantitative evidence across varying agricultural environments.
Comment 2: The article mentions several machine learning algorithms, but there is insufficient discussion on their adaptability to different crops and environmental conditions. The authors should further analyze the generalization capability of these models and propose solutions for improving adaptability across varied agricultural environments.
Response: Good observation from the reviewer. We have now expanded section 4.3 which focuses on the use of machine learning models for irrigation scheduling while considering different data types and sources. Please read from L542 – 601.
Comment 3: Although the importance of interoperability and standardization among IoT devices is acknowledged, the article does not provide detailed technical solutions.
Response: We thank the review for this comment. We have now revised section 5.1 and it now provides implementation guidance for MQTT and CoAP protocols. Additionally, Table 5 now illustrates practical deployment scenarios across irrigation contexts. Therefore, this technical framework bridges theoretical interoperability requirements with real-world implementation considerations.
Comment 4: Network security risks in IoT-based irrigation systems are mentioned, but the article lacks an in-depth exploration of potential threats and solutions.
Response: We agree with the reviewer, and we thank him/her for this observation. In addressing this issue, we have now added the security framework in Table 6 which explains these specific threats, mitigation strategies, and also provides a practical security roadmap while maintaining system accessibility.
Comment 5: The article highlights the importance of data preprocessing and quality control but fails to provide specific technical details.
Response: Thank you for this comment. We have now revised table 4 and it demonstrates how adaptive thresholding and sensor fusion algorithms address real-world data quality challenges in irrigation deployments.
Comment 6: The cost-benefit analysis of smart irrigation systems is somewhat limited, particularly in terms of their suitability for small-scale farmers.
Response: This is a valuable observation, and we have now incorporated economic analyses across different scales of implementing smart irrigation systems. Please read L993 – 1043 where challenges of using such systems in small-scale farmers are also highlighted.
Comment 7: Although cloud computing and edge computing are mentioned, the article does not explore the potential bottlenecks in data processing.
Response: We thank the reviewer for this comment. We have now revised section 3.2 (L403 – 483) where we explain the bottlenecks and their solutions.
Comment 8: While different irrigation methods are mentioned, the article does not provide a detailed comparison of their applicability in automated systems.
Response: We appreciate the reviewer’s observation. Therefore, we have revised Section 4.3.1 and modified Figure 6 where the automated systems are compared as well as their application capability while acknowledging practical implementation constraints.
Comment 9: The article touches on the role of decision support systems but does not explain how they function or are implemented.
Response: Thank you for the comment. Therefore, we have expanded Section 4.3 to illustrate the complete pathway from sensor data to irrigation decisions, grounded in real-world implementations that demonstrate how these systems function in practice.